# Exploration of the Moderating Effects of Physical Activity and Early Life Stress on the Relation between Brain-Derived Neurotrophic Factor (BDNF) rs6265 Variants and Depressive Symptoms among Adolescents

**DOI:** 10.3390/genes13071236

**Published:** 2022-07-13

**Authors:** Catalina Torres Soler, Sofia H. Kanders, Susanne Olofsdotter, Sofia Vadlin, Cecilia Åslund, Kent W. Nilsson

**Affiliations:** 1Centre for Clinical Research, Region Västmanland, Uppsala University, 72189 Västerås, Sweden; catalina.torres.soler@regionvastmanland.se (C.T.S.); susanne.olofsdotter@regionvastmanland.se (S.O.); sofia.vadlin@regionvastmanland.se (S.V.); cecilia.aslund@regionvastmanland.se (C.Å.); kent.nilsson@regionvastmanland.se (K.W.N.); 2Department of Psychology, Uppsala University, 75142 Uppsala, Sweden; 3Department of Public Health and Caring Sciences, Uppsala University, 75122 Uppsala, Sweden; 4The School of Health, Care and Social Welfare, Mälardalen University, 72123 Västerås, Sweden

**Keywords:** BDNF, moderation, depression, physical activity, stress

## Abstract

Depression affects one in five persons at 18 years of age. Allele A of the brain-derived neurotrophic factor (*BDNF)* rs6265 is considered to be a risk factor for depression. Previous studies of the interaction between *BDNF* rs6265, early adversity, and/or physical activity have shown mixed results. In this study, we explored the relation between *BDNF* rs6265 polymorphism and childhood stress, as well as the moderating effect of physical activity in relation to depressive symptoms using binary logistic regressions and process models 1, 2 and 3 applied to data obtained at three times (waves 1, 2 and 3) from the Survey of Adolescent Life in Västmanland cohort study (SALVe). Results revealed that both childhood stress and physical activity had a moderation effect; physical activity in wave 1 with an *R*^2^ change = 0.006, *p* = 0.013, and the Johnson–Neyman regions of significance (RoS) below 1.259, *p* = 0.05 for 11.97%; childhood stress in wave 2 with the *R*^2^ change = 0.008, *p* = 0 002, and RoS below 1.561 with 26.71% and >4.515 with 18.20%; and a three-way interaction in wave 1 in genotype AA carriers. These results suggest that allele A is susceptible to physical activity (positive environment) and childhood stress (negative environment).

## 1. Introduction

Depression is a common disease and an important cause of disability [1]. Its prevalence increases with age from childhood to adulthood [2,3,4,5], with higher increases in females [6]. The rate of adolescent depression is estimated to be one in five at 18 years of age according to previous studies [6,7]. Depression generates serious consequences in more than one area, including relationship problems with family members and peers, as well as poor school function. The development of symptoms leads to comorbidity and, in the worst cases, suicide [8,9,10,11]. Despite efforts to enhance early detection and treatment, results have been poor, and knowledge improvement in more effective primary interventions is required [12,13]. 

In 2017, Liu proposed some factors related to depression [14]. The author included candidate genes, such as brain-derived neurotrophic factor (*BDNF*), serotonin transporter polymorphisms (*5-HTTLPR*) [15], stress sensitivity, and neuroendocrinological process regulation response to stress [16] as mediators of depression risk and cognitive styles [17,18], interpersonal factors [19], early adversities [20], and neurobiological factors (such as changes in hippocampus and amygdala [16]) as moderating mechanisms of risk.

The protein called brain-derived neurotrophic factor (BDNF) is extensive in the brain and has a modulation function in neurotransmission, plasticity and learning [21,22,23]. The *BDNF* gene, which encodes BDNF, is located on chromosome 11p13, position 27,658,369 [24,25]. A variant of single nucleotide polymorphism (SNP) named Val66Met or rs6265 has been associated with depressive symptoms [26]. Allele G encodes Valine (Val), and allele A encodes Methionine (Met) [27]. *BDNF* rs6265 alleles are hereinafter referred to as G and A. There are different genotype frequencies among population groups. For instance, rates of AA polymorphism vary from 0% in the Maasai group in Kinyawa, Kenya, to 25.5% in the Chinese population in Beijing, China, while GG polymorphism varies from 98.1% in the Maasai in Kinyawa, Kenya, to 27.7% in the Chinese in Beijing, China. In Italians, the frequencies are 5.9% for AA, 36.3% for GA, and 57.8% for GG [27]. The mutation of G to A in the *BDNF* rs6265 decreases the regulated secretion of BDNF, which would modify synaptic modulation [28]. Carriers of allele A would have lower levels of BDNF for plasticity in their development [23]. The study of developmental follow-up models is important because the available data suggest an attenuation of the effect of genotypic differences with age [23,29,30]. This would allow for the identification of time periods sensitive for preventive interventions [23]. 

BDNF has been related to depressive symptoms, among others, because low levels of BDNF have been found in patients with major depression [29]. In animal models, when stress induces depression, a downregulation of BDNF has been observed [30]. In addition, allele A could be considered a risk factor for suicidal behaviour in depressed patients [27]. The results regarding the effect on depressive symptoms have been inconsistent, indicating that there are still factors to be examined that moderate the influence of BDNF on the risk of depression [31].

Childhood adversity has been shown to predict higher levels of depressive symptoms among adults, with a greater impact on genotype GA and AA carriers than GG carriers [32]. In addition, childhood adversity is associated with the severity, course and recurrence of depression and with a lack of response to treatment [33,34,35]. Kessler et al. estimated the risk proportion of adverse experiences in childhood for mood disorders to be 22.9% [36]. The mechanisms by which adverse experiences affect depression are not yet fully understood [37]. Some genes, such as *5-HTTLPR*, *CRHR1* and *FKBP5*, have been associated with susceptibility to stress [38,39,40,41]. In addition, it is known that stress reduces BDNF levels in humans [42]. Those who at least have one allele A appear to have an elevated cortisol response [43]. Some studies have shown an interaction effect of *BDNF* rs6265 with stressful life events but not with childhood adversity [44]. On the other hand, significant interactions have been observed between *BDNF* rs6265 and self-reports of childhood adversity in individuals with impaired hippocampal growth and depression [45].

Physical activity has a protective impact on stress tolerance, and the promotion of physical health has been considered one factor of acquired resilience [46,47]. In adolescents, training for 30 min twice a week for 10 weeks showed significant reductions in stress and depression [48]. There is a positive relation between levels of exercise and levels of BDNF [49,50,51,52]; acute aerobic exercise results in a temporary increase in circulating BDNF [51,52,53]. In a meta-analysis from 2016 [54], resting peripheral BDNF concentrations were significantly higher after exercise intervention >2 weeks. No effect was found regarding the duration of exercise intervention, exercise session time, and the number of exercise sessions per week [54]. According to [55], it is “the aerobic energy expense” that drives the impact on *BDNF* gene expression at a metabolic level. The temporary increase in the level of this protein is possibly explained by upregulated cellular processing [53]. On the other hand, the results of the relation between BDNF depressive symptoms and physical activity have been contradictory. Physical activity has been shown to protect girls with allele A [56], while a study of midlife subjects concluded that the *BDNF* rs6265 polymorphism did not amplify or mitigate the effect of physical activity on depressive symptoms [31]. Further research in a cohort of adolescents in the Netherlands did not show that the *BDNF* genotype modified the association between physical activity and depressive symptoms [57].

It has been reported that stress has a negative effect on physical activity and that physical activity has a positive effect on reducing stress [58]. Higher levels of cortisol have been found in A carriers [59], suggesting a negative relation between BDNF protein and stress response [60]. Longitudinal research is required to study the interaction mechanism of early adverse experiences, physical activity and neurobiological risk factors. 

Our study was based on four main premises: (a) *BDNF* polymorphism regulates the expression of BDNF protein [24], and high levels of BDNF are supposed to have a protective effect against neurodegeneration in the brain [61,62]; (b) physical activity temporarily increases BDNF expression and improves mood [63,64]; (c) early stress leads to neurological adaptations that increase the risk of depression, and *BDNF* polymorphism apparently moderates the occurrence of depressive symptoms in adults who have had adverse experiences in childhood [34,65,66]; and (d) the differences in effects appear to be associated with specific periods in life [23,67].

The results of previous studies on the interaction between *BDNF* genotypes and physical activity on depressive symptoms are contradictory [33,58,59], and other studies have separated the effects of stress during childhood in relation to candidate genes and their relationship to depressive symptoms [68].

The aim of our study was to explore the interaction between *BDNF* rs6265 polymorphism and childhood stress in relation to depressive symptoms, as well as the possible moderating effect of physical activity, in a cohort of adolescents. 

## 2. Materials and Methods

### 2.1. Participants

The study sample was derived from the Survey of Adolescent Life in Västmanland cohort study (SALVe cohort). In 2012, all adolescents born in 1997 and in 1999 who were living in the county of Västmanland, Sweden, were considered for the study (N = 5233). After excluding individuals living in Sweden for less than five years (*n* = 358), the adolescents and parents were contacted by regular mail with an invitation to take part in the study. After excluding individuals with mental disability, severe illness, and language difficulties, 4712 adolescents were eligible to participate in the study. The first assessment in 2012 (wave 1) included a self-report questionnaire, a saliva self-collection kit for deoxyribonucleic acid (DNA) extraction, and an informed consent form. The second assessment in 2015 (wave 2) and the third assessment in 2018 (wave 3) included self-report questionnaires. The adolescents with available data on all the variables of interest were included in this study. The study sample is presented in detail in Table 1. All participants signed an informed consent form in wave 1. Additional consent was also obtained from the legal guardian for participants younger than 15 years at the start of the study. Ethical approval was obtained from the Ethical Review Board of Uppsala (Dnr 2012/187). The study was performed in accordance with the Declaration of Helsinki.

### 2.2. Measures

Saliva samples were obtained through a self-collection kit (Oragene DNA, DNA Genotek, Ottawa, ON, Canada) in wave 1. The genotyping rate for all eligible study participants was 86.5%. DNA extraction was performed using a silica-based method (Kleargene™, LGC, Biosearch Technologies, Berlin, Germany) from 200 μL of saliva. The genotyping of *BDNF* rs6265 (val66met) polymorphism (G/A) variants was achieved using KASP™ and described as GG = 0, GA = 1 and AA = 2 (LGC, Biosearch Technologies). 

The assessment of depressive symptoms during the previous two weeks was conducted using the Depression Self-Rating Scale (DSRS) [65] at all three time points. The total number of symptom criteria (range: 0–9), based on the DSM-IV A criterion for major depression, was calculated into a continuous variable, “number of depressive symptoms”. The participants were coded as “showing depressive symptoms” in a binary variable if at least one of the general criteria and four more symptom criteria were present, as done in previous research [66]. The variables were created identically for all three time points.

The neuropattern pre-/postnatal stress questionnaire (NPQ-PSQ) is a tool designed to detect and treat stress pathology [69,70]. The Swedish version used is a translation made by the SALVe research group following recommended procedures [71,72] and adapted by adding more age levels. In wave 2, the parent indicated whether the adolescent had experienced (“No” or “Yes”) 19 different stressful events including occurrences such as accidents, physical or sexual abuse, time in hospital, emotional neglect, and time in foster care during their childhood/upbringing. If they responded “Yes”, they then indicated the age at which the event was experienced (0–1, 2–5, 6–10, 11–15 or “unknown”). In an additional question, the parents were also instructed to consider all these stressful events together and indicate the overall level of stress during childhood ranging from no stress (1) to high stress (10). This scale was used as an indicator of “childhood stress” in a continuous variable [69]. 

The participants self-reported physical activity associated with increased breathing/sweating during leisure time lasting more than 30 min. The seven options were “every day”, “4–6 times a week”, “2–3 times a week”, “once a week”, “1–3 times a month”, “less than once a month” and “never”. The answers were coded into a continuous variable ranging from 0 to 6, with “0” for “never” up to 6 for “every day”. Variables for physical activity were created in the same manner for all three time points.

Age was calculated using the date the form was sent out and the date of birth. The participants were coded as male (1) or female (2) using the embedded information in the social security number regarding the assigned birth sex. 

### 2.3. Statistical Analyses

Crude analyses of sex differences were performed for age, number of depressive symptoms, showing depressive symptoms, physical activity and *BDNF* rs6265 using the χ^2^ test for categorical variables and the Mann–Whitney U test for continuous variables. 

The moderating effect of childhood stress and/or physical activity on the relation between *BDNF* rs6265 and the continuous variable “number of depressive symptoms” and between *BDNF* rs6265 and the binary variable “showing depressive symptoms” were analysed in a step-wise manner using separate analyses for all three time points (Figure 1). 

Models of moderating effects are represented in Figure 1 [73]. The model for the conditional effect of *BDNF* rs6265 (X) on the number of depressive symptoms moderated (Y) by childhood stress (W) and analogously for the conditional effect of *BDNF* rs6265 on number of depressive symptoms moderated by physical activity (W) corresponds to Figure 1a. The conditional effect of *BDNF* rs6265 on the number of depressive symptoms moderated by both childhood stress (W) and physical activity (Z) corresponds to Figure 1b. The conditional effect of *BDNF* rs6265 on the number of depressive symptoms moderated by physical activity (Z) as a moderator of the childhood stress moderator (W) is modelled in the three-way interaction in Figure 1c. These analyses were performed using the Process macro tool (version 3.5 [73]). In addition, binary logistic regression analyses were performed to further assess these interaction effects (Figure 1) using “showing depressive symptoms” as the outcome for all three time points. Age, sex and physical activity/childhood stress were included as covariates. The Statistical Package for the Social Sciences (SPSS, version 27, Armonk, NY, USA) was used for the statistical analyses. 

## 3. Results

### 3.1. Crude Analyses

Females presented a higher mean number of depressive symptoms than males, and significantly more females than males showed depressive symptoms at each of the three measurement times. No statistically significant differences in the distribution of *BDNF* rs6265 alleles were found between the sexes or the different age groups. In wave 1 and wave 2, mean childhood stress was higher in females, whereas physical activity was higher in males at all times (Table 1). Chi square tests for independence indicated no significant association between *BDNF* rs6265 alleles and depression (considered non-significant if *p* > 0.05): Wave 1, χ^2^ (2, 1337) = 0.725, *p* = 0.7, and phi = 0.02; wave 2, χ^2^ (2, 1269) = 2.01, *p* = 0.33, and phi = 0.42; wave 3, χ^2^ (2, 890) = 4.58, *p* = 0.1, and phi = 0.07. 

### 3.2. Interactions

#### 3.2.1. One-Way Interactions

To explore the possible interactions between *BDNF* rs6265 and physical activity and between *BDNF* rs6265 and childhood stress, some binary logistic regressions were performed in all three waves using depressive symptoms at the time of the measure as the outcome and sex in wave 1, age at the time of the measure, and *BDNF* rs6265 and physical activity or childhood stress as variables in the binary regressions. Analysed models with significant interactions are shown in Table 2.

In wave 1, Model A1 had χ^2^ (5, 1337) = 52.56, and *p* < 0.001 and Model A2 had χ^2^ (7, 1337) = 57.87, and *p* < 0.001. The models explained 39–74% and 42–82% of the variance of showing depressive symptoms, respectively, and both correctly classified 88.1% of the cases. These results demonstrate statistically significant interaction between *BDNF* rs6265 GA and physical activity.

In wave 2, Model B1 had χ^2^ (5, 1269) = 115.83, and *p* < 0.001 and Model B2 had χ^2^ (7, 1269) = 130.43, and *p* < 0.001. The models explained 8.7–13% and 9.8–14.5% of the variance of depressive symptoms, respectively, in wave 2, and both correctly identified 75.6% of cases. Model B2 showed a statistically significant interaction between *BDNF* rs6265 AA and childhood stress.

Additionally, binary logistic regressions using similar models were performed to explore the importance of the relationship between (a) childhood stress and showing depressive symptoms in wave 1, (b) physical activity and showing depressive symptoms in wave 2, (c) childhood stress and showing depressive symptoms in wave 3, and (d) physical activity and showing depressive symptoms in wave 3. No statistically significant effects were found.

In brief, the results of the models in binary logistic regressions indicated that there were interactions between *BDNF* rs6265 GA and physical activity in wave 1 and between *BDNF* rs6265 AA and childhood stress in wave 2.

In wave 1, a moderation analysis (Figure 1a, Model 1) was performed to investigate the effect of physical activity as a simple moderator [73]. The outcome variable was the number of depressive symptoms, the predictor variable was *BDNF* rs6265 (GG, GA and AA), and the covariates were age and sex. The effect was significant (*R*^2^ = 0.076, *p* < 0.001), and the addition of interaction showed a significant change (*R*^2^ = 0.006, *p* = 0.013). The interaction between *BDNF* rs6265 GA and physical activity was significant, with *b* = 0.224, 95% CI (0.069, 0.379), and *p* = 0.005 for those with genotype *BDNF* rs6265 GA at a low moderator value (physical activity 2). For those with a middle moderator value (physical activity 4), the conditional effect of physical activity was *b =* −0.45, 95% CI (−0.811, −0.089), and *p* = 0.036, and for those with a high moderator value (physical activity 5), the effects of physical activity were not significant. Figure 2 shows the frequency of the physical activity on the horizontal axis, the number of depressive symptoms on the vertical axis, and three lines for *BDNF* rs6265 polymorphism with the continuous line for GG, the dashed line for GA, and the line with a series of points for AA. The figure shows that an increase in physical activity is accompanied by a reduction in the number of presented depressive symptoms. Although the slope is steepest for the AA carriers, the reduction in the number of depressive symptoms for GA carriers was the only statistically significant effect. Additionally, the slopes of the lines representing the *BDNF* rs6265 polymorphisms indicate that the largest differences in the moderation effect were found at level two (1–3 times a month) of physical activity.

The Johnson–Neyman region of significance for the conditional effect of *BDNF* rs6265 on number of depressive symptoms at different levels of physical activity was found to be below 1.259, *p* = 0.05 for 11.97%; see Figure 3.

In wave 2, Model 1 was used to investigate the moderation effect of childhood stress on the relation between the predictor *BDNF* rs6265 and the outcome depressive symptoms in wave 2. Sex in wave 1 and age in wave 2 were used as covariates. *R*^2^ was 0.164, *p* < 0.001, and the addition of the interaction made a significant change (*R*^2^ = 0.008, *p* = 0.002). Statistically significant interactions were found between *BDNF* rs6265 AA and childhood stress (*b* = 0.776, 95% CI (0.3, 1.253), and *p* = 0.001 and between rs6265 GA childhood stress (*b* = 0.153, 95% CI (0.009, 0.296), and *p* = 0.037). 

Moreover, the conditional effect of *BDFN* rs6265 AA on the number of depressive symptoms at a low moderator value (childhood stress 1) was −1.236, 95% CI (−2.263, −0.209), and *p* = 0.018. At a middle moderator value (childhood stress 2), there was no significant effect, and at a high moderator value (childhood stress 5), the conditional effect was 1.869, 95% CI (0.435, 3.302), and *p* = 0.011. These results indicate a positive effect of the relationship of childhood stress and number of depressive symptoms; see Figure 4. 

Figure 4 shows childhood stress level on the horizontal axis, the number of depressive symptoms on the vertical axis, and three lines for *BDNF* rs6265 polymorphism with the continuous line for rs6265 GG, the dashed line for rs6265 GA, and the line with a series of points for rs6265 AA. When childhood stress level was low (1), rs6265 AA genotype carriers showed the lowest number of depressive symptoms, and when childhood stress was at the highest level (5), carriers of the rs6265 AA genotype presented the highest number of depressive symptoms. The increase in the number of depressive symptoms with childhood stress was less pronounced in rs6265 GA and rs6265 GG carriers, indicating susceptibility to childhood stress in individuals with the rs6265 AA polymorphism.

The Johnson–Neyman regions of significance for the conditional effect of *BDNF* rs6265 on depressive symptoms at different levels of childhood stress were below 1.561 with 26.71% and above 4.515 with 18.20%; see Figure 5.

#### 3.2.2. Two-Way Interactions 

Childhood stress and physical activity

In wave 2, a binary logistic regression was performed to examine the influence of the interactions between *BDNF* rs6265 and childhood stress and between *BDNF* rs6265 and physical activity on depressive symptoms; see Table 3.

In Table 3, Model C included sex, age in wave 2, physical activity in wave 2, childhood stress, and *BDNF* rs6265 variants. In Model C2, interactions between *BDNF* rs6265 and physical activity and between *BDNF* 6265 and childhood stress were added. For Model C2, χ^2^ (8, 1269) = 140.99, *p* < 0.001. The overall correct prediction was 76% with statistically significant predictors of sex, physical activity and childhood stress, as well as the interactions between *BDNF* rs6265 AA and childhood stress with an OR 4, 95% CI (1.11–14.46). The binary logistic regression analyses of the two-way interactions in waves 1 and 3 were not significant.

Moderation analysis using Model 2 (see Figure 1b) with the number of depressive symptoms as the outcome was also performed for wave 2, with *BDNF* rs6265 as the predictor and childhood stress (W) and physical activity (Z) as moderators. The model summary *R*^2^ was 0.182, *p* < 0.001, and the addition for the interaction between *BDNF* rs6265 and childhood stress showed a statistically significant change (*R*^2^ = 0.008, *p* = 0.003). The combined interactions between *BDNF* rs6265 and childhood stress and between *BDNF* rs6265 and physical activity resulted in *R*^2^ = 0.009, *p* = 0.006. The interaction effect for *BDNF* rs6265 GA and childhood stress was −1.62, *t* (1269) = 2.229, 95% CI (0.019, 0.305), and *p* = 0.026, and the interaction effect for *BDNF* rs6265 AA and childhood stress was 7.11, *t* (1269) = 2.840, 95% CI (0.220, 1.203), and *p* = 0.005. The conditional effect of the focal predictor *BDNF* rs6265 AA was 1.996, 95% CI (0.399, 3.593), and *p* = 0.014 at moderator levels of 5 for childhood stress and 2 for physical activity, as well as 1.761, 95% CI (0.298, 3.224), and *p* = 0.018 at moderator levels of 5 for childhood stress and 4 for physical activity. At other moderator levels, no significant results were found.

The respective analyses in waves 1 and 3 did not show significant interactions for both predictors.

#### 3.2.3. Three-Way Interactions 

The binary logistic regressions used to study the three-way interactions were not statistically significant in any wave, even though the analysis in wave 1 using moderation Model 3 (see Figure 1c) with the outcome variable of the number of depressive symptoms in wave 1, the predictor variable *BDNF* rs6265, and moderators of childhood stress (W) and physical activity in wave 1 (Z) showed a significant result at *R*^2^ = 0.348, *p* < 0.001, and the addition of the interaction had a significant effect at *R*^2^ = 0.004, *p* = 0.038. In addition, the interaction between *BDNF* rs6265 AA, childhood stress and physical activity in wave 1 was found to be significant, *b* = −0.323, 95% CI (−0.594, −0.051), and *p* = 0.011; see Figure 6.

Figure 6 compares the interactions of the *BDNF* rs6265 GG, GA and AA genotypes with childhood stress and physical activity and their relationship with the number of depressive symptoms. The horizontal axis shows physical activity, the vertical axis shows the number of depressive symptoms, and three lines show the level of childhood stress (the continuous line for childhood stress = 1, the series of points line for childhood stress = 2 and the dashed line for childhood stress = 5). Figure 6c shows that for carriers of the AA genotype who had experienced a high level of childhood stress (5), increased physical activity reduced the number of depressive symptoms. This effect was reduced at a moderate stress level (2), and no interaction was observed at the low stress level (1). In Figure 6a,b for the *BDNF* rs6265 GG and GA genotypes, respectively, the reduction in depressive symptoms as an interaction effect of childhood stress and physical activity shows a similar trend for different levels of childhood stress.

The results of the test of highest order of unconditional interactions using Process analyses in the three waves are summarized in Table 4. 

In wave 1, statistically significant interactions were found between *BDNF* rs6265 and physical activity using Models 1 and 2. Model 3 showed a significant three-way interaction for *BDNF* rs6265 AA, physical activity and childhood stress (see Figure 4). In wave 2, interactions between *BDNF* rs6265 and childhood stress were found using Models 1 and 2.

In brief, the results obtained by applying the different analysis methods were congruent. The results showed a sensitivity of *BDNF* rs6265 GA and AA carriers to the beneficial effects of physical activity with a reduction in the number of depressive symptoms at the wave 1 time point. They also indicated a higher sensitivity for *BDNF* rs6265 AA carriers to present an increased number of depressive symptoms in wave 2 when exposed to increased levels of stress during childhood.

The visual representation of the three-way interactions in wave 1 indicates that for the *BDNF* rs6265 AA carriers who were exposed to higher stress, the increase in physical activity had a protective effect with respect to the presence of depressive symptoms.

The absence of significant correlations in wave 3 may be related to the reduction in the number of participants reducing the measurability of the effect or a reduction in the protective effect of physical activity in older adolescents.

## 4. Discussion

The results of this study demonstrate statistically significant interactions between *BDNF* rs6265 and physical activity as a positive environmental factor on the prevention of depressive symptoms in wave 1. In turn, the relationship between *BDNF* rs6265 and childhood stress as a negative environmental factor and the effect on the number of depressive symptoms was also statistically significant in wave 2. In addition, a significant three-way interaction was found in wave 1. The complementary study of moderation models supported the significance of these results, showing the moderation effect of the positive and negative environmental factors that suggests the plasticity characteristic of rs6265 A allele.

Varied results have been reported regarding the interaction between *BDNF* rs6265 and environmental factors. The *BDNF* rs6265 A allele has been considered a risk for the development of depression [32,45] because allele A carriers have shown a low activity-regulated expression of BDNF and the alteration of hippocampal activation [74,75]. Furthermore, the interaction between exposure to childhood stress and being an allele A carrier becomes a predictor of the development of depressive symptoms [44]. However, increased levels of BDNF have a protective effect against depression [76]. Former studies have shown that 30 min of physical activity (endurance ride) is sufficient to transiently elevate BDNF levels [49,50]. When examined, interaction effects for carriers of allele A and moderate physical activity as a protective effect against development of depressive symptoms were found [56].

In this study, the exploration of the moderating effect of *BDNF* rs6265 on two environmental factors (physical activity, with an alleged protective effect, and childhood stress as a negative factor) in relation to the development of depressive symptoms allowed us to study the differential susceptibility of *BDNF* rs6265 alleles [77,78,79]. 

The results indicate the existence of a moderation effect of physical activity in GA carriers in wave 1, which is in line with the results obtained by Mata et al. in 2010 [56]. Additionally, in wave 1, when the combined interaction of *BDNF* rs6265 polymorphism, childhood stress, and physical activity was studied, it became apparent that there was a three-way interaction for the AA carriers. When AA carriers were exposed to high levels of childhood stress, they experienced a reduction in depressive symptoms if their physical activity increased. This effect was greater than for AA carriers who experienced lower levels of childhood stress.

The observed moderation effect of childhood stress in AA carriers in wave 2 is consistent with the effect on depression described by Gatt et al. in 2009 [26].

We therefore conclude that AA carriers have a susceptibility to present depression symptoms at the time of wave 2 if they have been exposed to a high level of childhood stress, contrary to Stavrakakis et al.’s claims [57]. The negative effect of an increased number of depressive symptoms related to experiencing high levels of childhood stress appears be a long-term effect, as it was found in adolescents in wave 2. 

The absence of statistically significant interactions in wave 3 is more difficult to interpret because it may have been due to the reduction in the number of participants.

However, frequent physical activity prevents the negative consequence of depressive symptoms [63]. Additionally, the same positive effect of the reduction in depressive symptoms was found in *BDNF* rs6265 GA carriers. 

The crude analysis results are in accordance with those of previous studies. The proportion of individuals with depression and depressive symptoms, as well as the reported levels of childhood stress, were significantly higher among females [80], and the frequency of physical activity was higher in males [81]. No differences in the frequency of *BDNF* rs6265 polymorphism by sex were found, and the frequencies of alleles were similar to those found in an Italian sample [27].

The exploration of interactions using binary logistic regressions also indicated that there is a significant interaction between the higher frequency of physical activity and the lower frequency of depressive symptoms in individuals with the *BDNF* rs6265 GA polymorphism, which represented about 30% of the wave 1 population. It is important to highlight that AA carriers only comprised about 3% of the same group, which could have affected the statistical significance in the analysis. 

Analog regression analyses performed in wave 2 showed statistically significant interactions between high levels of childhood stress and the *BDNF* rs6265 AA genotype, suggesting a predisposition of these individuals to develop depressive symptoms when exposed to high levels of childhood stress, as described in Hosang et al.’s systematic review in 2014 [68].

Furthermore, in wave 2, when both interactions were included in the model, the interactions of the GA genotype with physical activity and the AA genotype with childhood stress were also significant, showing comparable patterns of relationships to those obtained with models that were used to separately study interactions.

The Model 1, 2 and 3 results are congruent with those obtained with the binary logistic regressions, giving small but statistically significant additions to the change. Process analysis allowed for the graphic representation of the moderating effect considering the levels of the studied variables and defining the levels of significance of each moderator [73]. The relatively small percentage of significance levels may have been related to the lower frequency of presence of the allele involved in the interaction, and, in the case of physical activity, to the fact that many individuals performed physical activity at a moderate frequency and not daily. 

The found moderation effects may be related to a combination of multiple events related to the biology of BDNF, as summarized by Hing et al. [64]. First, BDNF is expressed by glutamatergic neurons [82] as a pre-pro-peptide in the beginning, later splits to pro-BDNF, and is then converted into BDNF. BDNF mediates several neuroplasticity processes [83], such as GABAergic neurons and GABAergic transmission [84]. However, at the cellular level, the effects of pro-BDNF and BDNF can be opposed (as apoptosis vs. the promotion of cell survival or the attenuation of neurogenesis vs. neurogenesis promotion in the hippocampus, respectively) [64]. Second, in SNP rs6265, the change to methionine modifies the binding site, impeding hippocampal synaptic activity [85]. Third, the relation of allele A to childhood adversity, which moderates the development of depressive symptoms, can be modified by interactions with other genes, such as the serotonin transporter SLC6A4 [86,87]. Fourth, in animal models, the effects of stress on BDNF expression in the brain show changes in neuroplasticity, with deficits in neurogenesis in the hippocampus and GABAergic activity [88,89] and a reduction in the tissue plasminogen activator that reduces the conversion of pro-BDNF to BDNF [90]. These results suggest that negative stressors mediate the effects of pro-BDNF expression [64]. Fifth, through epigenetic mechanisms, such as DNA methylation, negative stress changes the expression of BDNF [91] or the histone modification process, leading to a reduced expression of BDNF [92]. This suggests that the positive and negative effects overlap at different levels. 

It is important to address that the vast majority of studies investigating the Val66- and Met66-variant differences at the protein level have used animal models or cell cultures [93]. Results from a human study indicated that metabolic stress downregulates the expression of BDNF but not concentrations of plasma BDNF [93]. An alteration in the pro-BDNF/BDNF ratio deriving from the rs6265 polymorphism might be important to consider [94]. A complete overview of this complex system is out of scope for this paper.

The multiple interactions, with their temporal relationship, type of effect, and defined population, involve a number of methodological problems [95]. One methodological issue was carrying out multiple tests. There were 60 tests performed, giving a Bonferroni *p*-value of 0.00083, which left the statistical significance levels of all the interactions above those of the corrected *p*-value. However, the purpose of the study was exploratory, so the obtained results should be verified by further studies. We decided that the use of a corrected *p*-value, such the Bonferroni *p*-value, should not have been used in this study due to the type of study, the risk to increase the Type II error, and the sensitivity conferred on the adjustment by the relatively small size of rs6265 polymorphism AA group and the small effect [96,97,98,99].

When analysing the moderating effect of childhood physical stress on depressive symptoms within the studied intervals and RoS, the effect of differential susceptibility became evident. However, the effect of physical activity is less conclusive because of the reduced RoS in those who are less active [100]. In any case, it is important to consider the simultaneous effect of the positive influence of physical activity on the reduction in depressive symptoms for the same sensitivity factor, being an A allele carrier, because of its characteristics of susceptibility [101]. 

The study’s limitations include that the number of participants progressively reduced in the second and third measurements and that self-assessment questionnaires were used for reporting depressive symptoms and the level of physical activity. Information about childhood stress was collected from parents in wave 2 using an overall score on a scale without categorizing the events. It is possible that parents of adolescents with depressive symptoms may have a memory bias about the level of childhood stress experienced by their children.

In addition, sex and age were controlled in the models; however, other factors that might modify the presence of depressive symptoms such as antidepressant medication or cognitive behavioural therapy were not studied. Due to the low frequency of the *BDNF* rs6265 AA allele, it is difficult to interpret the magnitude of the results regarding its susceptibility. Moreover, due to the design of the study, it was not possible to quantify biomarkers such as BDNF concentration.

The study’s strengths include the opportunity to repeat measurements in a cohort, which allowed us to study the effects over time, particularly the effect of physical activity on inferred BDNF production and the windows for effect during development in adolescence. Additionally, the stability of the frequencies of sex and *BDNF* rs6265 polymorphism in the population facilitated the study of relationships with the other variables.

## 5. Conclusions

Physical activity and childhood stress have been shown to exert moderating effects on the relation between *BDNF* rs6265 polymorphism and depressive symptoms among adolescent carriers of GA and AA, respectively. These carriers have a reduced number of depressive symptoms when physical activity increases and childhood stress levels are low, but symptom numbers increase when childhood stress levels are high. Moreover, physical activity moderates the effect of childhood stress on the presence of depressive symptoms in carriers of the *BDNF* rs6265 AA polymorphism. Overall, the obtained results suggest that allele A confers differential susceptibility on childhood stress and the interactions of *BDNF* rs6265, childhood stress, and physical activity. This implies that mild increased physical activity exerts a preventive action for the occurrence of depressive symptoms in individuals with this genetic susceptibility who have experienced high levels of childhood stress. Given the exploratory nature of the current study, these results should be confirmed in another study. 

## Figures and Tables

**Figure 1 genes-13-01236-f001:**
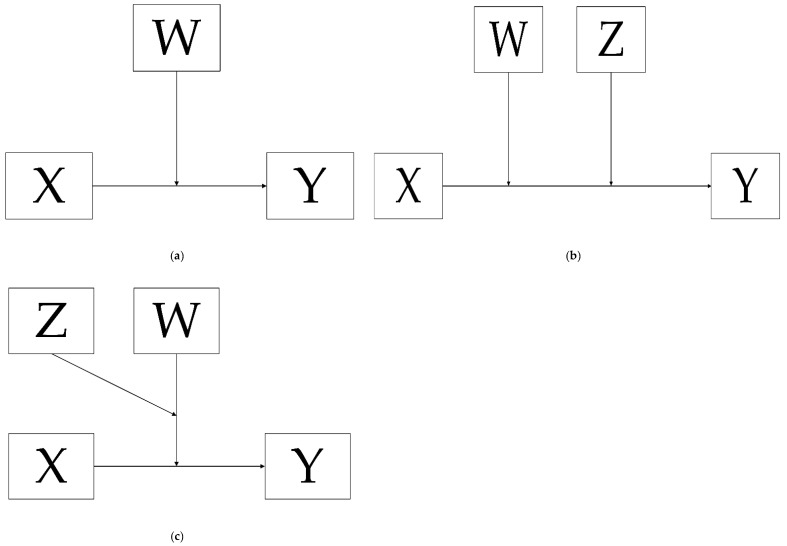
Conceptual diagram of three different moderation models, where X = *BDNF* rs6265 and Y = depressive symptoms. In addition, (**a**) W = childhood stress and physical activity; (**b**,**c**) W = childhood stress and Z = physical activity.

**Figure 2 genes-13-01236-f002:**
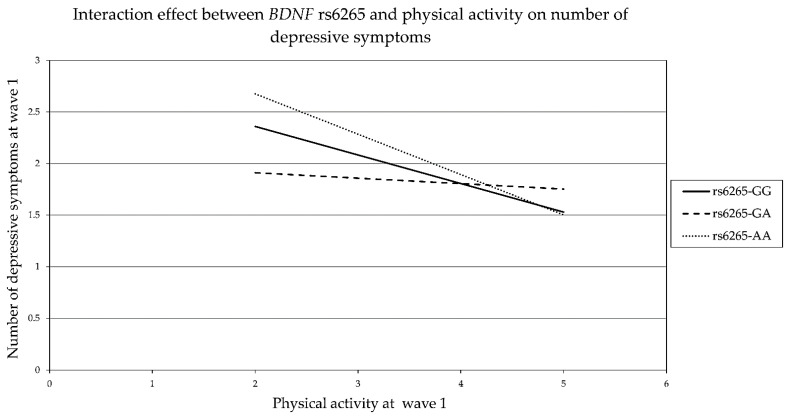
Interaction between BDNF rs6265 and physical activity in wave 1 (Model 1). Lines in the figure represent obtained data of the conditional effect of BDNF rs6265 GG, GA and AA at different levels of physical activity on the number of depressive symptoms.

**Figure 3 genes-13-01236-f003:**
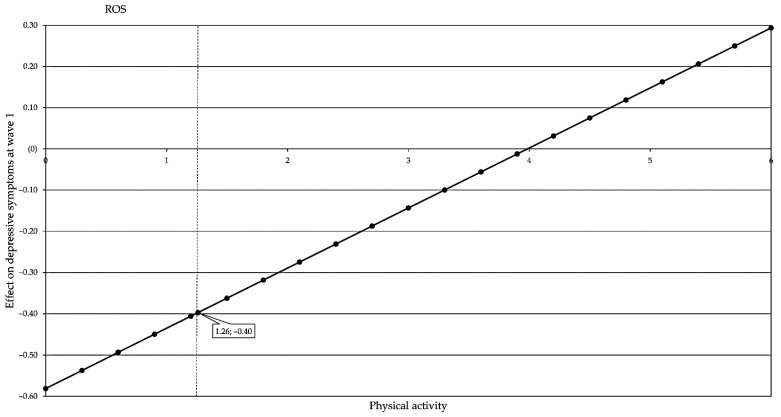
Region of significance (RoS) for the conditional effect of *BDNF* rs6265 on number of depressive symptoms in wave 1 at different levels of physical activity.

**Figure 4 genes-13-01236-f004:**
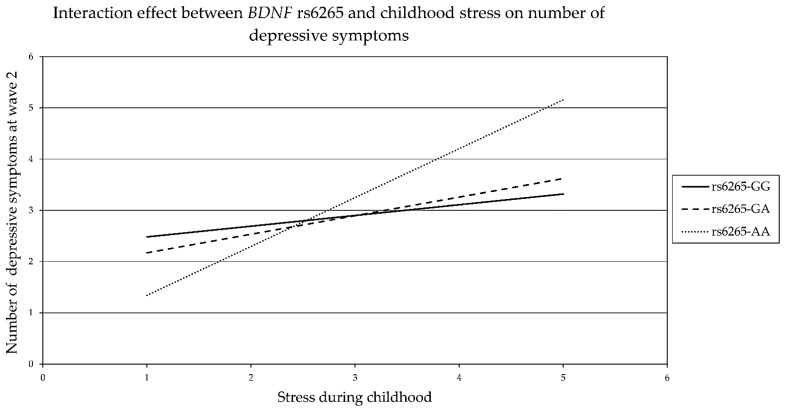
Interaction between *BDNF* rs6265 and childhood stress in wave 2 (Model 1).

**Figure 5 genes-13-01236-f005:**
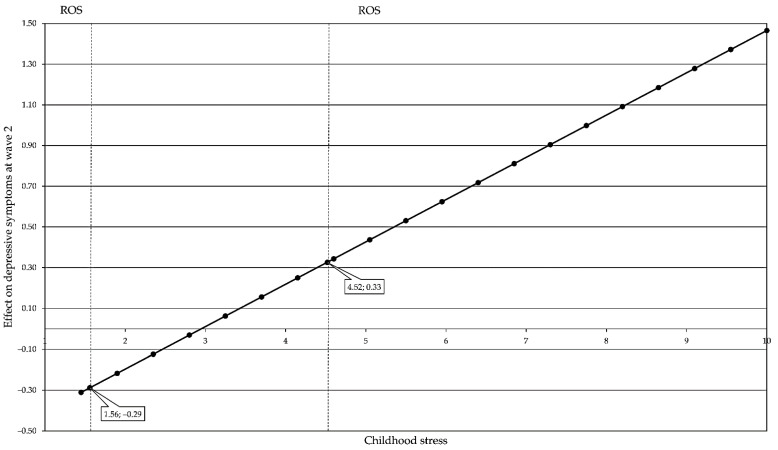
Region of significance (RoS) for the conditional effect of *BDNF* rs6265 on depressive symptoms in wave 2 at different levels of childhood stress.

**Figure 6 genes-13-01236-f006:**
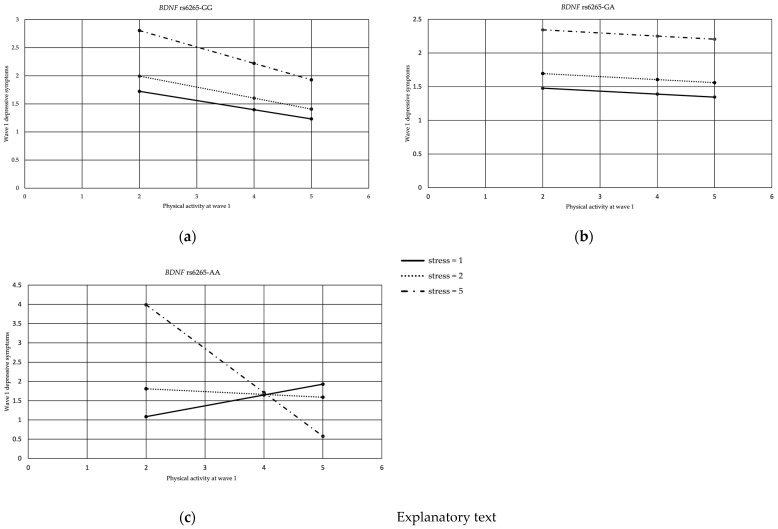
Interaction between *BDNF* rs6265, physical activity and childhood stress at different levels of stress in wave 1 in Model 3. (**a**) *BDNF* rs6265 GG, (**b**) *BDNF* rs6265 GA and (**c**) *BDNF* rs6265 AA.

**Table 1 genes-13-01236-t001:** Descriptive statistics for the study sample.

	Wave 1 (*n* = 1337)			Wave 2 (*n* = 1269)			Wave 3 (*n* = 890)		
	Female (*n* = 761)	Male (*n* = 576)	χ^2^/Z (*p*)	Female (*n* = 740)	Male (*n* = 529)	χ^2^/Z (*p*)	Female (*n* = 566)	Male (*n* = 324)	χ^2^/Z (*p*)
Age, years ^a^ (SD)	14.37 (1.029)	14.42 (1.038)	–0.197 (0.844)	17.30 (1.029)	17.35 (1.040)	–0.356 (0.721)	20.407 (1.0289)	20.425 (1.037)	–0.108 (0.914)
Number of depressive symptoms, mean (SD)	2.188 (2.350)	1.405 (1.807)	**–6.081** **(<0.001)**	3.57 (2.605)	1.90 (2.194)	**–11.840** **(<0.001)**	3.66 (2.840)	2.25 (2.564)	**–7.452** **(<0.001)**
Participants with depression (%)	123 (16.2)	36 (6.3)	30.747 (<0.001)	244 (33.0)	70 (13.2)	**64.555** **(<0.001)**	196 (34.6)	60 (18.5)	**26.100** **(<0.001)**
Childhood stress ^a^ (SD)				3.05 (2.046)	2.72 (1.817)	**–2.727** **(0.006)**			
Participants with workout > 30 min (%)			**–2.345** **(0.019)**			**33.396** **(<0.001)**			**35.211** **(<0.001)**
Never	51 (6.7)	49 (3.7)		57 (7.7)	47 (8.9)		63 (11.1)	28 (8.6)	
Less than once a month	36 (4.7)	24 (4.2)		54 (7.3)	27 (33.3)		108 (19.1)	43 (13.3)	
1–3 times a month	53 (7.0)	31 (5.4)		67 (9.1)	32 (6.0)		90 (15.9)	39 (12.0)	
Once a week	89 (11.7)	50 (8.7)		86 (11.6)	51 (9.6)		65 (11.5)	26 (8.0)	
2–3 times per week	280 (36.8)	185 (32.1)		250 (33.8)	139 (26.3)		146 (25.8)	81 (25.0)	
4–6 times per week	213 (28.0)	196 (34.0)		168 (22.7)	152 (28.7)		78 (13.8)	84 (25.9)	
Every day	39 (5.1)	41 (7.1)		58 (7.8)	81 (15.3)		16 (7.1)	23 (7.1)	
*BDNF* rs6265 alleles (%)			2.235 (0.327)			2.207 (0.332)			0.538 (0.764)
G: G	507 (66.6)	401 (69.6)		491 (66.4)	364 (68.8)		367 (64.8)	217 (67.0)	
G: A	232 (30.5)	155 (26.9)		228 (30.8)	147 (27.8)		181 (32.0)	96 (29.6)	
A: A	22 (2.9)	20 (3.5)		21 (2.8)	18 (3.4)		18 (3.3)	11 (3.2)	

The χ^2^ test was used for categorical variables, and the Mann–Whitney *U* test was used for continuous variables. ^a^ Analysed continuously. Significant results are shown in bold font.

**Table 2 genes-13-01236-t002:** Binary logistic regressions used to analyse the relationship between showing depressive symptoms and sex, age, and *BDNF* rs6265 variants and physical activity or childhood stress.

Wave 1 (*n* = 1337)	Model A1		Model A2	
	*p*	OR (95% CI)	*p*	OR (95% CI)
Sex in wave 1	**<0.001**	2.95 (1.99–4.36)	**<0.001**	2.89 (1.95–4.29)
Age in wave 1	**0.003**	1.29 (1.09–1.52)	**0.002**	1.3 (1.1–1.53)
Physical activity in wave 1	**0.003**	0.86 (0.77–0.95)	**<0.001**	0.8 (0.71–0.9)
*BDNF* rs6265 GG	0.44		0.05	
*BDNF* rs6265 GA	0.32	0.82 (0.56–1.21)	**0.02**	0.33 (0.13–0.84)
*BDNF* rs6265 AA	**0.49**	1.37 (0.55–3.41)	0.056	1.87 (0.23–15.14)
*BDNF* rs6265 GG × physical activity in wave 1			0.08	
*BDNF* rs6265 GA × physical activity in wave 1			**0.03**	1.3 (1.1–1.53)
*BDNF* rs6265 AA × physical activity in wave 1			0.076	0.92 (0.53–1.59)
Constant	**<0.001**		**<0.001**	
Wave 2 (*n* = 1269)	Model B1		Model B2	
	*p*	OR (95% CI)	*p*	OR (95% CI)
Sex in wave 1	**<0.001**	3.13 (2.32–4.22)	**<0.001**	3.13 (2.32–4.23)
Age in wave 2	0.13	1.11 (0.97–1.26)	0.11	1.11 (0.98–1.27)
Childhood stress	**<0.001**	1.24 (1.16–1.32)	**<0.001**	1.17 (1.08–1.26)
*BDNF* rs6265 GG	0.41		**0.01**	
*BDNF* rs6265 GA	0.39	0.88 (0.65–1.18)	0.05	0.57 (0.33–0.99)
*BDNF* rs6265 AA	0.27	0.6 (0.24–1.49)	**0.02**	0.01 (0–.44)
*BDNF* rs6265 GG × childhood stress			**0.02**	
*BDNF* rs6265 GA × childhood stress			0.07	1.14 (0.99–1.31)
*BDNF* rs6265 AA × childhood stress			**0.02**	3.87 (1.22–12.31)
Constant	**<0.001**		**<0.001**	

Models using variables of sex in wave 1 (male as reference) and *BDNF* rs6265 variants (GG as reference) in all regressions and showing depressive symptoms in wave 1 (1) or in wave 2 (2) as the outcome. Model A1: sex, age in wave 1, physical activity in wave 1, and *BDNF* rs6265 variants. Model A2: all the variables included in Model A1 with the addition of an interaction term *BDNF* by physical activity in wave 1. Model B1: sex, age in wave 2, childhood stress summary (parental report), and *BDNF* rs6265. Model B2: all the variables included in Model B1 with the addition of an interaction term *BDNF* rs6265 by childhood stress summary (parental report). Significant results are shown in bold font.

**Table 3 genes-13-01236-t003:** Binary logistic regressions exploring the influence of the interactions between *BDNF* rs6265 and physical activity and between *BDNF* rs6265 and childhood stress on depressive symptoms in wave 2.

	Model C			Model C2		
	*p*	OR	(95% CI)	*p*	OR	(95% CI)
Sex in wave 1	**<0.001**	**0.33**	**(0.24–0.44)**	**<0.001**	**3.04**	**(2.25–4.12)**
Age	0.28	1.08	**(0.94–1.23)**	0.23	1.08	**(0.95–1.24)**
Physical activity	**0.001**	**0.87**	**(0.81–0.95)**	**0.001**	**0.86**	**(0.78–0.94)**
Childhood stress	**<0.001**	**1.22**	**(1.15–1.30)**	**<0.001**	**1.16**	**(1.07–1.25)**
*BDNF* rs6265 GG	0.46			0.052		
*BDNF* rs6265 GA	0.30	1.63	(0.65–4.11)	0.07	0.45	(0.19–1.06)
*BDNF* rs6265 AA	0.44	1.45	(0.57–3.72)	0.10	0	(0–2.73)
rs6265 GG × physical activity				0.719		
rs6265 GA × physical activity				0.469	1.07	(0.87–1.27)
rs6265 AA × physical activity				0.687	1.22	(0.46–3.25)
rs6265 GG × stress				**0.020**		
rs6265 GA × stress				0.057	1.15	(1.00–1.32)
rs6265 AA × stress				**0.034**	**4.00**	**(1.11–14.46)**
Constant	0.04	0.08		**0.011**	**0.04**	

Binary logistic regressions using depressive symptoms in wave 2 as the outcome (*n* = 1269). Model C: sex (male as reference), age in wave 2, physical activity in wave 2 and *BDNF* rs6265 variants (GG as reference). Model C2 additionally includes interaction terms *BDRF* rs6265 × physical activity and *BDNF* rs6265 × stress. Significant results are shown in bold font.

**Table 4 genes-13-01236-t004:** Test of highest order of unconditional interactions using Process.

Outcome	Interaction Model	Interaction Term	*R*^2^ Change	*F*	*p*
Depressive symptoms wave 1	1	x*physical activity wave 1	0.006	4.345	**0.013**
	1	x*childhood stress	0.000	0.171	0.843
	2	x*childhood stress	0.000	0.018	0.982
		x*physical activity wave 1	0.004	3.142	**0.044**
		x*childhood stress and x*physical activity wave 1	0.004	1.593	0.174
	3	x*physical activity wave 1*childhood stress	0.004	3.290	**0.038**
Depressive symptoms wave 2	1	x*physical activity wave 2	0.002	1.479	0.228
	1	x*childhood stress	0.009	6.690	**0.001**
	2	x*childhood stress	0.008	5.973	**0.003**
		x*physical activity wave 2	0.001	0.830	0.436
		x*childhood stress and x*physical activity wave 2	0.009	3.690	**0.006**
	3	x*physical activity wave 2*childhood stress	0.003	1.966	0.141
Depressive symptoms wave 3	1	x*physical activity wave 3	0.003	1.555	0.212
	1	x*childhood stress	0.002	1.053	0.349
	2	x*childhood stress	0.003	1.705	0.182
	2	x*physical activity wave 3	0.005	2.330	0.098
		x*childhood stress and x*physical activity wave 3	0.007	1.700	0.148
	3	x*physical activity wave 3 *childhood stress	0.001	0.472	0.624

Interaction analyses using Process including age and sex as covariates and predictor x = *BDNF* rs6265. Model 1 = simple moderation; Model 2 = two moderators; Model 3 = three-way interactions. Model 1 used moderators of physical activity or childhood stress. Models 2 and 3 combined childhood stress (W) and physical activity (Z). *p* < 0.05 is reported in bold. * = ×.

## Data Availability

The data are not publicly available for confidentiality reasons.

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
