# Peer review of "Exploration of the Moderating Effects of Physical Activity and Early Life Stress on the Relation between Brain-Derived Neurotrophic Factor (BDNF) rs6265 Variants and Depressive Symptoms among Adolescents"

_genes, 2022, doi:10.3390/genes13071236_

Round 1

Reviewer 1 Report

I understand that the reviewed paper refers to a Retrospective cohort in which it was observed an interaction between the BDNF Val66met polymorphism and the development of depressive symptoms in adolescents, and these interactions were moderated by physical activity.

Statistical analysis suggests that individuals carrying the methionine allele were more sensible to childhood stress – as referring to the number of depressive symptoms –  as retrospectively observed from Wave 2 to Wave 1. It also suggests that homozygote individuals are more responsive to physical activity, regarding improvements in depressive symptoms.

The Retrospective Cohort has been well performed. Methods are clear and accurate. Statistics analyses are adequate. This is a relevant study for the scientific front.

BDNF is a neurotrophin mainly expressed in neural tissue with several roles in neural survival and maintenance, being essential for the proper function of neurons. The transcripts of BDNF gene leave the nucleus as a pre-pro-BDNF mRNA length which reaches the first steps of translation as a pro-BDNF tRNA length which forms the pro-BDNF isoform of protein that undergoes further cleavage systems (or controls) before release as BDNF. Such “controls” are expected to be more accurate in specific phases of the brain’s development (such as childhood and late in adolescence – brain maturation), regarding that pro-BDNF and BDNF signal through different dichotomic receptors, for cell death and growth respectively. So that, there is a fine and multilevel control in the release/secretion of BDNF whereof even memory (long-term potentiation) depends. The polymorphism in BDNF alters the genetic code of the ‘pro’ region of ‘pro-BDNF’.

So two main concerns arise: a) Relevant genetic changes in the BDNF gene are incompatible to life; b) Slight genetic changes in the BDNF gene should infer in impairment.

Bearing that in mind, I would like to make a few points that I hope to benefit the introduction and discussion:

I missed references on associations of the Val66met polymorphism and stress, as it has been pointed in the -   recent literature – I suggest: de Assis, G. G., & Gasanov, E. V. (2019). BDNF and Cortisol integrative system – Plasticity vs. degeneration: Implications of the Val66Met polymorphism.

Lines 88 – 90: “although no prolonged effects have been found”

Long-term/persistent effects of physical activity on basal BDNF levels have been reported in qualitative and qualitative synthesis, see: Dinoff, A., et al. (2016). The Effect of exercise training on resting concentrations of peripheral brain-derived neurotrophic factor (BDNF): A meta-analysis. De Assis, G. G., et al. (2018). Brain derived neutrophic factor, a link of aerobic metabolism to neuroplasticity.

In Table 2 (1. Wave 1 and 2. Wave 2): I suggest to keep a standard nomenclature for the BDNF genotypes throughout the whole text, figures, and tables, and revising “BDNF (1) and BDNF(2)”.

In discussion, I strongly suggest enriching the text with the complexity of the multilevel systems influencing BDNF release (and therefore signaling), which has been recently demonstrated in humans, consider consulting:

de Assis, G. G., et al. (2021). The Val66 and Met66 Alleles-Specific Expression of BDNF in Human Muscle and Their Metabolic Responsivity.

de Assis, G. G., & Hoffman, J. R. (2022). The BDNF Val66Met Polymorphism is a Relevant, But not Determinant, Risk Factor in the Etiology of Neuropsychiatric Disorders – Current Advances in Human Studies: A Systematic Review.

Overall, I think this study was carefully handled by the pairs and brings high-quality Observational analysis to the field of behavioral genomics and the preventive potential of epigenetics. Glad to read it.

Author Response

Thank you for your comments

Reviewer 2 Report

This paper is well written and presents a cohort study conducted among the Survey of Adolescent Life in Västmanland (SALVe) individuals. Statistical analysis performed using ctest and Mann–Whitney U test indicating the higher proportion of females with depression and depressive symptoms is interesting. Exploration of interactions between physical activity, childhood stress and BDNF rs6265 polymorphism using binary logistic regressions provides more insights about the influence of various factors that reduce depression.

While the topic is interesting and worth further study, there are few minor concerns:

1. Is there a set of control population that can be compared with the individuals taken for the cohort study?

2. Is the cohort group (SALVe) considered in this study are the people who lived in Sweden for more than five years at the time of wave 1 assessment in 2012?

3. In the methods sections, while I appreciate the description of Crude analysis in the Statistical analysis section in more detail, a more descriptive details of logistic regression and how it is applied to the dataset of study here would be of more interest.

4. Discussions on the limitations of the study can be improved in the perspective of the study design based mainly on the self-assessment questionnaires. Is only 3% of the cohorts have AA polymorphism as mentioned in the discussion section “It is important to 503 highlight that AA carriers comprised only about 3% of the same group, which could have 504 affected the statistical significance in the analysis.” A more detailed description on AA carriers would provide more insights about the polymorphism among these cohorts. 

5. Is there a history of medication among these cohorts? Does it influence the reduction in depression other than physical activity?

Author Response

Thank you for your comments
